# TRAINING-FREE SELF-SCHEDULING FOR EFFICIENT LLM INFERENCE SERVING

## ABSTRACT

The ability to deliver fast responses under strict latency requirements is critical for Large Language Model (LLM) inference serving. Most existing systems rely on a first-come-first-served (FCFS) scheduling policy, which often suffers from head-of-line blocking. While a number of solutions have been proposed, they typically require training additional models or auxiliary predictors, such as BERT, to estimate decoding lengths. These approaches limit generalization and necessitate retraining for new domains or distributions. To address these limitations, we propose self-scheduling with LLM, a novel approach that leverages the reasoning capabilities of the LLM itself without requiring extra training or auxiliary models. We systematically investigate a range of feasible strategies and conduct extensive analyses. Experimental results show that our method achieves up to a $5\times$ improvement in TTFT, a $3\times$ improvement in TPOT, a $6\times$ reduction in latency, and a $9\times$ increase in throughput under both general and domain-specific workloads, with negligible overhead. This work offers a lightweight yet intelligent scheduling paradigm, demonstrating both practicality and strong potential for LLM inference serving.

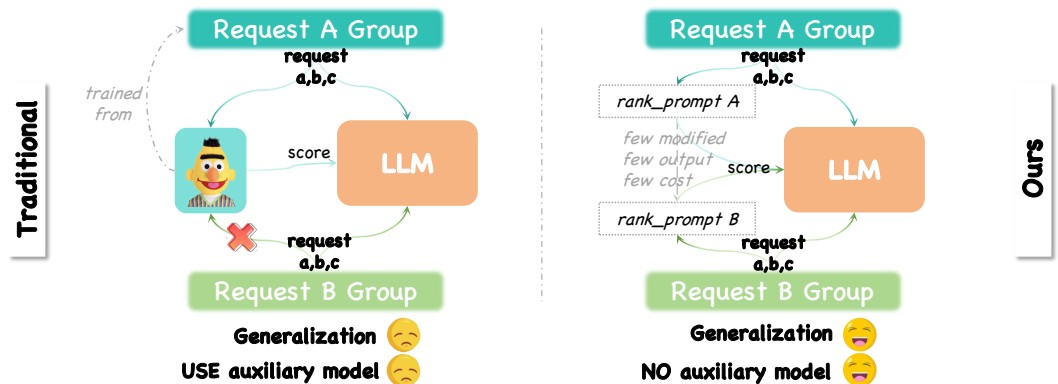

Figure 1: Comparison of traditional scheduling methods and the proposed approach.

## 1 INTRODUCTION

Large Language Models (LLMs) and their applications such as ChatGPT (OpenAI, 2025), DeepSeek (DeepSeek-AI et al., 2025), Qwen (Yang et al., 2025), Gemini (Google, 2025), and Cursor (Cursor, 2025) are increasingly integrated into daily life, processing massive volumes of user requests. This demand poses substantial challenges to meeting Service Level Objectives (SLOs), particularly with respect to maintaining low latency. Since users expect rapid responses, measuring and optimizing key performance metrics is essential. The primary metrics of concern are Time to First Token (TTFT), Time per Output Token (TPOT), and Total Latency (Agarwal et al., 2023; Zhong et al., 2024; Qin et al., 2024; Yu et al., 2022; Agrawal et al., 2024). Improving these metrics remains a central objective in LLM optimization research, with efficient scheduling for LLM inference emerging as a critical direction for addressing these challenges (Zhou et al., 2024; Zhen et al., 2025).

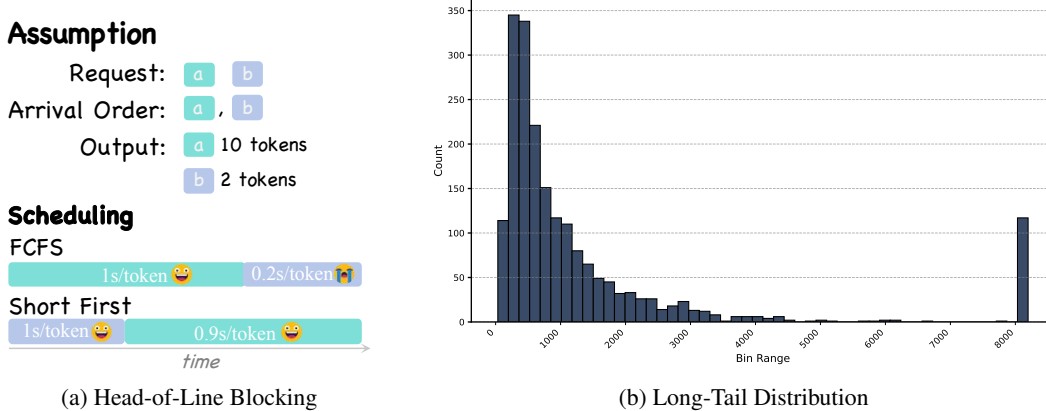

(a) Head-of-Line Blocking

(b) Long-Tail Distribution

Figure 2: Illustration of head-of-line blocking and real-world data distributions. (a) Prioritizing shorter requests improves overall efficiency. Here, the default assumption is that the LLM processes one token per second, with the prefill effect ignored. (b) Response length distribution of Qwen3-14B on a subset of NuminaMath dataset.

A widely adopted scheduling strategy is first-come-first-served (FCFS), which is simple to implement but suffers from head-of-line blocking (Kaffes et al., 2019) due to the long-tail phenomenon, as shown in Figure 2. In this setting, a few long requests may delay the execution of many shorter ones, resulting in severe inefficiencies. To address this challenge, an increasing body of research has explored predicting request lengths and scheduling accordingly (Cheng et al., 2024; Hu et al., 2024; Zheng et al., 2024b; Hua et al., 2025; Fu et al., 2024). These methods typically employ auxiliary models such as regression predictors, lightweight LMs, or embedding-based classifiers to estimate generation length prior to decoding. Although effective in controlled settings, such approaches face two key limitations. First, they introduce additional training overhead, thereby increasing system complexity. Second, their generalization capacity is limited: when request distributions or application domains shift, retraining or fine-tuning is often required, as illustrated in Figure 1. This lack of adaptability poses a significant barrier to real-world deployment.

LLMs, however, possess powerful reasoning capabilities, as demonstrated in tasks such as LLM-as-a-Judge (Li et al., 2025), which has been widely applied across diverse scenarios. Building on this observation, we propose a new paradigm: *Training-Free Self-Scheduling* for LLM inference. Rather than relying on external predictors, we directly leverage the reasoning ability of the LLM itself to assist in scheduling decisions. This approach eliminates the need for additional training or auxiliary models, making it lightweight, adaptive, and easy to deploy. We further introduce a self-scheduling-aware starvation mitigation mechanism to ensure fairness in cases where requests might otherwise be indefinitely delayed.

Extensive and fine-grained experimental analyses demonstrate that our method is both simple and effective: it generalizes seamlessly across domains and achieves substantial improvements in TTFT, TPOT, and overall latency. Beyond performance gains, our study offers a broader insight: LLMs can function not only as inference engines but also as intelligent schedulers of their own workloads. This work thus opens a new direction for intelligent scheduling in LLM systems.

To summarize, our contributions are as follows:

- **Training-Free Self-Scheduling**: We introduce a novel scheduling paradigm that eliminates the reliance on auxiliary models and retraining.

- **Strategy Design and Evaluation**: We systematically develop and assess multiple strategies for leveraging the reasoning capabilities of LLMs, thoroughly exploring the feasibility and effectiveness of self-scheduling.

- **Empirical Validation**: We demonstrate that our approach yields substantial improvements in TTFT, TPOT, and latency across diverse workloads, domains, and decoding modes.

- **Future Research Directions**: We present a practical and intelligent solution that not only addresses current challenges in LLM inference scheduling but also opens a promising direction for future research.

## 2 RELATED WORKS

While our work introduces LLM self-scheduling as a lightweight and adaptive paradigm, it is closely related to prior efforts on inference scheduling and generation length prediction. In this section, we review existing approaches, highlighting their strengths and limitations, and position our method in relation to two major directions: (a) LLM Inference Scheduling, (b) scheduling methods based on length prediction, and (c) the emerging paradigm of LLMs as self-evaluators.

### 2.1 LLM INFERENCE SCHEDULING

The rapid growth of LLM applications has heightened the need for efficient inference scheduling (Miao et al., 2023; Yuan et al., 2024; Zhou et al., 2024). Various approaches have been proposed to improve efficiency, such as iterative scheduling (Yu et al., 2022), along with a range of open-source frameworks, including vLLM (Kwon et al., 2023), TensorRT-LLM (NVIDIA, 2023), SGLang (Zheng et al., 2024a), TGI (HuggingFace, 2023), DeepSpeed-MII (Microsoft, 2022), and llama.cpp (ggml org, 2022). In most of these frameworks, FCFS remains the default scheduling policy, highlighting the necessity of exploring more efficient alternatives.

### 2.2 LENGTH PREDICTION APPROACHES

The uncertainty of generation length poses one of the greatest challenges in request scheduling, making accurate length prediction a fundamental requirement. Prior studies have proposed a variety of prediction techniques that can be broadly classified into three categories (Zhen et al., 2025).

*Exact prediction* methods estimate the token count directly using approaches such as BERT embeddings with random forest regression (Cheng et al., 2024), lightweight OPT models (Hu et al., 2024), or constrained regression techniques (Qiu et al., 2024b). *Range-based classification* methods instead partition requests into length bins, either by predicting ranges from prompts (Zheng et al., 2024b; Jin et al., 2023; Jain et al., 2024; Qiu et al., 2024a; Stojkovic et al., 2024; Hua et al., 2025) or by employing real-time classifiers over token embeddings (Shahout et al., 2024). *Relative ranking* methods focus on ordering requests according to expected lengths; for example, Fu et al. (2024) predicts pairwise relationships within the same batch to enhance robustness and mitigate overfitting.

Notably, Zheng et al. (2023) introduced the earliest attempt to leverage LLMs for directly predicting request lengths. However, at that time, LLMs lacked the strong instruction-following ability of current models and still required additional post-training to perform length prediction effectively.

They either rely on auxiliary models or require additional training, which imposes limitations on both generalizability and usability.

### 2.3 LLM AS A JUDGE AND SELF-EVALUATION

The paradigm of using LLMs as evaluators has become a widely adopted reward mechanism (Li et al., 2025). Typical approaches include assigning scores to individual responses (Li et al., 2024a; Xie et al., 2025), ranking multiple candidates (Li et al., 2024c), or selecting one or more options from a given set (Yao et al., 2023; Li et al., 2024b). In this work, we primarily adopt scoring and ranking strategies.

Overall, existing solutions reduce latency but remain constrained by training overhead and poor generalization. Inspired by LLM-as-a-Judge, our work is the first to systematically explore LLM self-scheduling, which leverages the model's own reasoning ability for scheduling without auxiliary predictors.

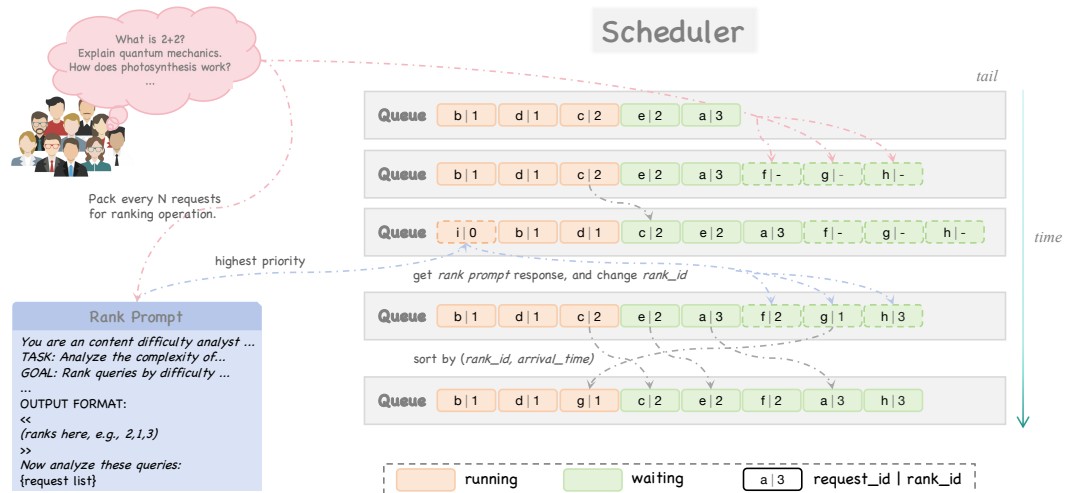

Figure 3: An illustrative example of a simple self-scheduling mechanism without starvation control.

## 3 METHODOLOGY

### 3.1 PROBLEM FORMULATION

We consider the problem of scheduling a set of requests $\mathcal{R} = \{r_1, r_2, \ldots, r_n\}$ submitted to an LLM inference server. Each request $r_i$ produces a response of unknown length $L(r_i)$ measured in tokens. Since the true lengths are not available prior to decoding, a scheduling policy must rely on estimates $\hat{L}(r_i)$ to decide the execution order. The goal is to minimize tail latency by avoiding head-of-line blocking caused by long requests.

Formally, the scheduling task can be expressed as learning a ranking function $f : \mathcal{R} \to \mathbb{R}^n$ that outputs a permutation over requests. The ideal ranking $R^*$ is obtained by sorting requests in ascending order of their true lengths $L(r_i)$.

### 3.2 LLM SELF-SCHEDULING

Instead of training auxiliary predictors, we leverage the reasoning ability of the LLM itself to estimate request characteristics. Given a set of requests, we prompt the LLM to provide a relative ranking of their expected response lengths. The LLM outputs a predicted rank $\hat{R}$, where ties are allowed (i.e., multiple requests may be assigned the same rank).

This approach is lightweight and adaptive: no additional training is required, and the same method can be applied across different domains without retraining.

### 3.3 EVALUATION METRIC: KENDALL'S TAU-B WITH TIES

Since the predicted ranking $\hat{R}$ may contain ties, we evaluate agreement with the gold ranking $R^\star$ using Kendall's tau-b:

$$\tau_b(R_a, R_b) = \frac{n_c - n_d}{\sqrt{(n_0 - n_a)(n_0 - n_b)}}, \tag{1}$$

where $n_c$ and $n_d$ are the numbers of concordant and discordant pairs, $n_0 = \binom{n}{2}$ is the total number of pairs, and $n_a$, $n_b$ are the numbers of tied pairs in $R_a$ and $R_b$. This extension ensures ties are handled fairly, making $\tau_b$ a robust measure for our setting.

Thus, $\tau_b$ ranges from $-1$ (completely reversed order) to $+1$ (perfect agreement), with *ties properly handled*.

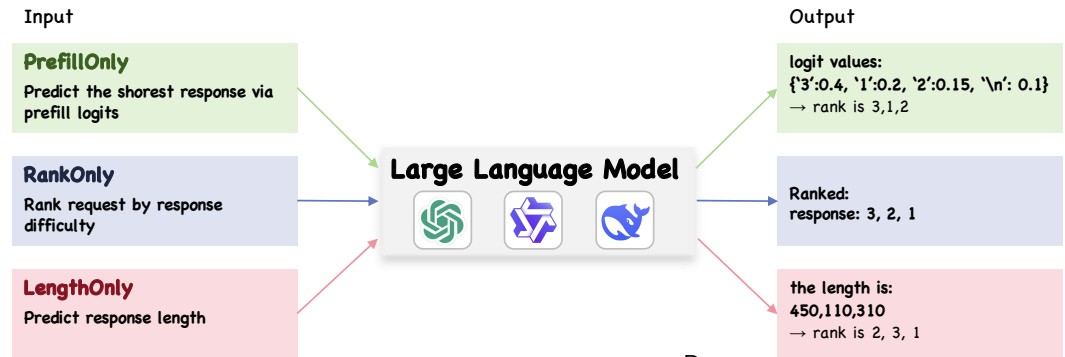

Figure 4: Overview of our LLM-based scheduling methods.

### 3.4 CONSTRUCTING THE GOLD RANKING WITH TIES

**Goal.** We seek a gold ranking $R^\star$ of requests by true response length that (i) orders requests from short to long and (ii) treats near-equal responses as ties. This is necessary because our evaluation metric (Kendall's $\tau_b$) explicitly accounts for ties, and because tiny length differences should not induce artificial strict orderings.

**Setup.** Let $\mathcal{R} = \{r_1, \dots, r_n\}$ be requests with true response lengths $L(r_i) \in \mathbb{R}_{\geq 0}$. We first sort requests in non-decreasing order of length:

$$(r_{(1)}, \dots, r_{(n)}) \quad \text{s.t.} \quad L(r_{(1)}) \leq \cdots \leq L(r_{(n)}).$$

For brevity, denote $L_{(i)} := L(r_{(i)})$.

**Tie rule (relative threshold).** For each adjacent pair $(i-1, i)$ with $i \geq 2$, we define a pairwise relative threshold:

$$\varepsilon_i \;=\; \alpha \times \frac{L_{(i-1)} + L_{(i)}}{2}. \tag{2}$$

We declare $r_{(i-1)}$ and $r_{(i)}$ tied if

$$|L_{(i)} - L_{(i-1)}| \;\leq\; \varepsilon_i. \tag{3}$$

When $L_{(i-1)} = L_{(i)} = 0$, we set $\varepsilon_i = 0$; hence zero-length items are tied only if exactly equal.

**Rank assignment.** Let $g_{(i)}$ be the gold rank of $r_{(i)}$. We use "competition ranking" (a.k.a. 1224 ranking): the first item gets rank 1. For $i \geq 2$,

$$g_{(i)} \;=\; \begin{cases} g_{(i-1)}, & \text{if } |L_{(i)} - L_{(i-1)}| \leq \varepsilon_i \quad \text{(tie)}, \\ i, & \text{otherwise.} \end{cases} \tag{4}$$

This yields ranks such as $1, 1, 3, 4, 4, 6, \dots$, preserving ties and skipping integers after tied groups.

### 3.5 LLM-BASED RANKING STRATEGIES

Our approach processes a batch of incoming requests (e.g., 10 requests) and uses the LLM to predict their relative execution order or expected response lengths. We propose three variants, as shown in Figure 4:

**PREFILLONLY**: Leverages the logits from the prefill stage. Since the token–probability mapping reflects the likelihood of each request being the shortest, the highest-probability logit corresponds to rank 1, the second highest to rank 2, and so on.

**RANKONLY**: Directly prompts the LLM to output a ranked list of request IDs (e.g., 1,4,3,5,2) with only a few decoding steps.

**LENGTHONLY**: Asks the LLM to predict the response length (in tokens) for each request in the batch, after which the requests are ranked according to the predicted lengths.

---

**Algorithm 1** Unified Batching and Priority Scheduling

---

**Require:** batch_size $B$, timeout $T$, threshold, quantum_steps, max_promotions
1: $Q \leftarrow \emptyset, \mathcal{R} \leftarrow \emptyset$, timer off
2: **while** running **do**
3:    **if** request arrives **then**
4:       init request and push to $Q$; start timer if off
5:    **end if**
6:    **if** $|Q| \geq B$ **or** timer$\geq T$ **then**
7:       make `rank_prompt` from $Q$ with `rank_id`$= -\infty$; add to $\mathcal{R}$; clear $Q$; reset timer
8:    **end if**
9:    // Anti-starvation promotion
10:   find and promote candidates with starvation $\geq$ threshold
11:   set priority $\leftarrow$ TRUE, quantum $\leftarrow$ quantum_steps for promoted requests
12:   sort $\mathcal{R}$ by (priority, rank_id, arrival_time); select runnable set $S$
13:   **for** each $x \in S$ **do**
14:      run($x$); $x$.starvation $\leftarrow 0$
15:      **if** $x$ is `rank_prompt` AND $x$ finishes **then**
16:         retrieve rank info from $x$'s response
17:         for each request in the original batch, update its `rank_id` based on the rank info
18:      **end if**
19:      **if** $x$.priority **then**
20:         $x$.quantum–; if $x$.quantum $\leq 0$, $x$.priority $\leftarrow$ FALSE
21:      **end if**
22:   **end for**
23:   **for** each $y \in \mathcal{R} \setminus S$ **do**
24:      $y$.starvation++
25:   **end for**
26:   remove finished from $\mathcal{R}$
27: **end while**

---

### 3.6 UNIFIED SELF-SCHEDULING WITH ANTI-STARVATION MECHANISM

To prevent requests with longer responses from being blocked for excessive periods, we integrate our self-scheduling approach with an anti-starvation mechanism, as illustrated in Algorithm 1.

## 4 EXPERIMENT

### 4.1 DATASET

To evaluate the robustness of our proposed methods, we selected a diverse set of datasets spanning both general-purpose and domain-specific tasks, including mathematics and code. These datasets also cover a wide spectrum of difficulty levels, ensuring that the evaluation reflects performance across varied and challenging scenarios.

**NuminaMath** (Beeching et al., 2024): This dataset is specifically designed to evaluate a model's mathematical reasoning abilities. It includes a vast collection of competition-level math problems accompanied by detailed chain-of-thought (CoT) solutions, which help enhance a model's step-by-step reasoning for complex tasks.

**TACO** (Li et al., 2023): Short for "Topics in Algorithmic COde generation dataset", this is a large-scale, open-source dataset designed to be a more challenging benchmark for code generation models. It features competition-level programming questions focused on algorithmic topics, which helps evaluate a model's understanding and reasoning in real-world programming scenarios.

**ShareGPT** (Team, 2023): This is a high-quality, open-source conversational dataset widely used for training LLMs on instruction-following and dialogue generation tasks. It comprises filtered conversation samples derived from real-world user interactions with advanced language models.

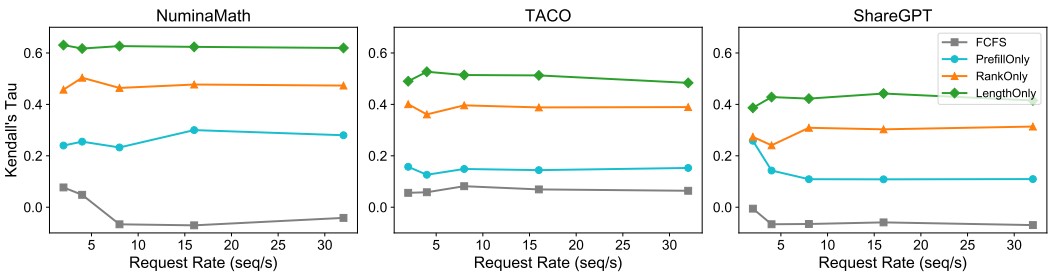

Figure 5: Kendall's Tau comparison of FCFS, PREFILLONLY, RANKONLY, and LENGTHONLY.

## 4.2 ENVIRONMENT

All experiments were conducted on a server equipped with eight NVIDIA A100 GPUs, each with 40 GB of memory. The software environment was configured with CUDA 12.8 and vLLM version 0.9.2. We employed vLLM version 0.9.2 with the following configuration: starvation_threshold_steps = 4, priority_quantum_steps = 32, and max_promotions_per_round = 16. To ensure the generalizability of our results, each experiment was repeated four times with different random seeds, and the reported values correspond to the average across these runs. With respect to preemption and swapping operations, we followed the default implementation provided in vLLM. In addition, we did not employ chunked prefill in our experiments.

## 5 ANALYSES

### 5.1 KENDALL'S TAU

As illustrated in Figure 5, Kendall's Tau varies across the three proposed approaches in different domains. the performance improves progressively from FCFS to PREFILLONLY, RANKONLY, and finally LENGTHONLY. This progression is logically consistent. PREFILLONLY relies on the probability distribution over the first token, which does not fully capture the model's generative behavior. RANKONLY implicitly leverages the model's ability to predict sequence length for ranking, but this indirect approach leads to some performance degradation. In contrast, LENGTHONLY directly predicts the decoding length and constructs the ranking offline, making it the most straightforward and effective strategy. Intuitively, mathematical problems are relatively easier to predict, conversation queries in ShareGPT are more complex, and code-related tasks fall in between.

The small fluctuations observed at lower request rates can be attributed to the reduced batch size in those settings. As the request rate increases, the batch size approaches its maximum (i.e., 10), stabilizing the performance trends.

In terms of computational overhead, PREFILLONLY incurs the least cost as it requires only prefilling, RANKONLY demands a moderate amount of decoding overhead ($\sim 20$ tokens), while LENGTHONLY has the highest decoding overhead ($\sim 40$ tokens). We provide additional experiments to validate this extra-cost analysis in Sec 5.4.

### 5.2 PERFORMANCE

For a comprehensive evaluation of our three methods (PREFILLONLY, RANKONLY, and LENGTH-ONLY), we report TTFT, TPOT, and latency per request at P95 across NuminaMath, TACO, and ShareGPT (Figure 6).

At request rates below 10 (Poisson Distribution), some fluctuations appear. This occurs because a larger proportion of small requests are batched with rank prompts, thereby increasing the number of ranking operations. In addition, sorting multiple requests together is more effective than handling them sporadically, as it introduces more comparative relationships.

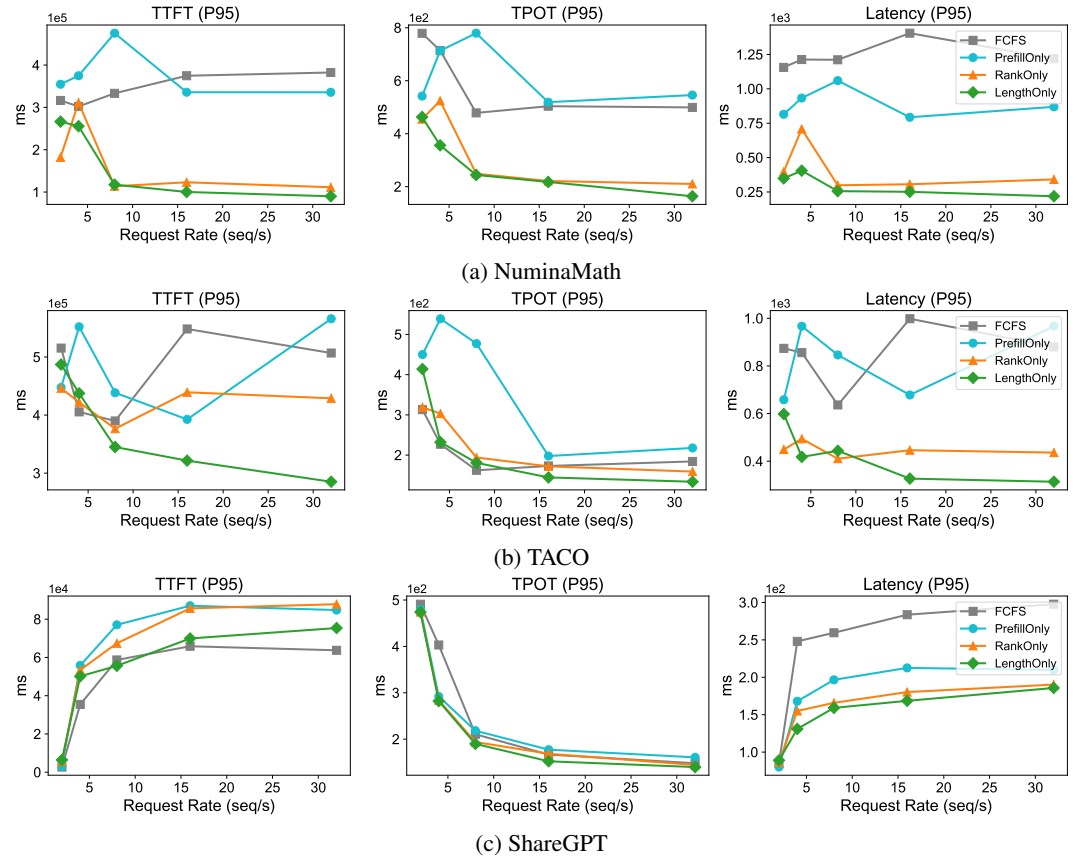

Figure 6: Performance of NuminaMath, TACO, and ShareGPT on TTFT, TPOT, and latency at P95.

As the request rate increases, performance gradually stabilizes. Among the proposed methods, LENGTHONLY consistently achieves the best results across all metrics. This observation aligns with Sec. 5.1, confirming that ranking accuracy is directly correlated with overall performance.

For TTFT, improvements are most pronounced on the math (5×) and code (1.6×) datasets but relatively modest on ShareGPT. This is because math and code exhibit stronger long-tail effects: longer requests can be deferred while shorter ones are prioritized, thereby reducing delays. In contrast, ShareGPT shows a weaker long-tail distribution, so reordering yields less benefit.

For TPOT, we also observe clear improvements, particularly in math is 2.5×. By grouping requests of similar lengths, reordering enhances GPU parallel utilization and mitigates the drag caused by long-tail requests, resulting in smoother generation.

Finally, latency shows the most substantial improvement, with math achieving 6×, Code 3×, and ShareGPT 2×. Short requests are completed more quickly, reducing delay for the majority of tasks, while longer requests are still efficiently processed in parallel, further improving GPU utilization.

We further evaluate the methods in thinking mode across all datasets (Figure 7), showing that response lengths in thinking and non-thinking modes remain relatively consistent. This demonstrates the generalization ability of our approach across both domains and decoding modes.

*Note that all reported results are obtained with ranking operations enabled.*

### 5.3 THROUGHPUT

To conduct a comprehensive throughput analysis, we evaluated each method using 3,000 requests under two experimental settings: (i) a fixed duration of 10 minutes and (ii) a fixed workload of 500 requests. Table 1 summarizes the throughput achieved by the baseline and the LENGTHONLY

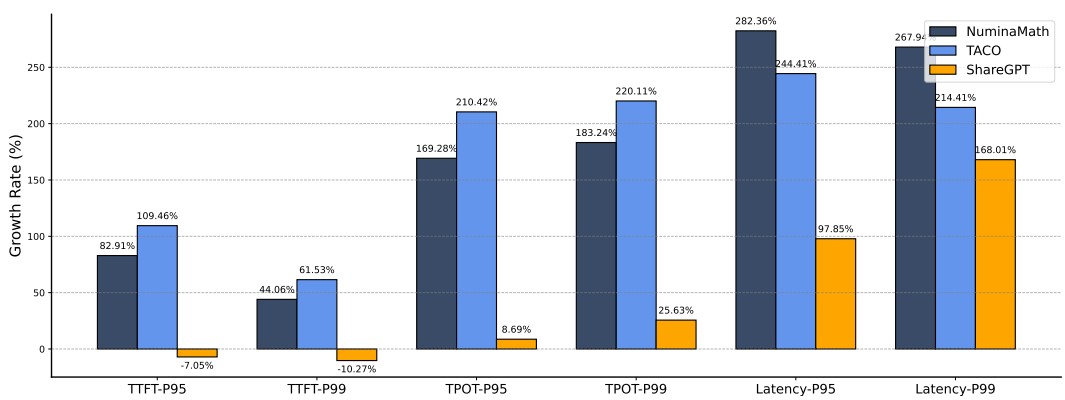

Figure 7: Growth rate comparison of NuminaMath, TACO, and ShareGPT in thinking mode.

| Dataset | Fixed Duration (10 minutes) | | | Fixed Workload (500 requests) | | |
|---|---|---|---|---|---|---|
| | FCFS | LENGTHONLY | Speedup | FCFS | LENGTHONLY | Speedup |
| NuminaMath | 211 | 948 | 4.49× | 28.50 min | 3.48 min | 9.33× |
| TACO | 202 | 412 | 2.04× | 38.27 min | 11.83 min | 3.23× |
| ShareGPT | 689 | 986 | 1.43× | 6.23 min | 4.03 min | 1.55× |

Table 1: Throughput comparison of scheduling methods across multiple domains.

strategy across multiple datasets. The results show consistent improvements, ranging from modest to substantial gains, confirming that our method benefits not only end users but also service providers.

## 5.4 EXTRA COST

To evaluate the worst-case scenario, we select 20 math requests with rate = 64 and batch size = 1, since larger batches naturally mitigate this effect. In this setting, the system requires two ranking prompts and a fixed decoding length

| Dataset | FCFS | LENGTHONLY |
|---|---|---|
| NuminaMath | 11m 6s 530ms | 11m 6s 585ms |

Table 2: Extra cost in 20 requests.

of 40 tokens. As shown in Figure 2, LENGTHONLY introduces only 55 ms of additional latency compared with the baseline, demonstrating that our method incurs negligible overhead.

## 5.5 STARVATION

Our rank-aware starvation prevention method significantly improves performance. We conducted an experiment comparing our method with a no-starvation method. The results demonstrate a remarkable improvement: a nearly 50% reduction in TPOT-P99 and a nearly 30% reduction in Latency-P99. Importantly, these improvements were achieved without compromising performance on other P99 metrics, which remained consistent.

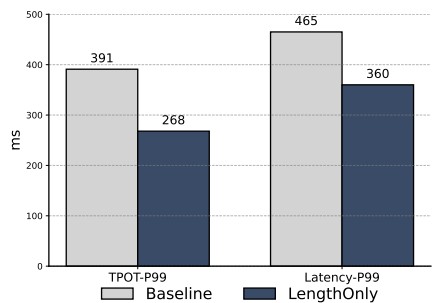

Figure 8: Starvation ablation.

## 6 CONCLUSION

This work is the first to demonstrate the feasibility of leveraging existing LLMs for self-scheduling without training additional models. We proposed three scheduling strategies, from decoding-free to minimally decoding approaches, and showed their effectiveness across diverse domains, including mathematics, code, and general-purpose datasets, under both thinking and non-thinking modes. To further mitigate starvation, we incorporated a rank-aware prevention mechanism. Our study introduces a novel paradigm that opens new directions for intelligent scheduling in LLM inference.

## 7 ETHICS STATEMENT

This work focuses on improving the efficiency of LLM inference serving through self-scheduling techniques. Our contributions are primarily methodological and system-level, without introducing new datasets or collecting human subjects data. All datasets used in this study (NuminaMath, TACO, and ShareGPT) are publicly available and widely adopted in prior research.

## 8 REPRODUCIBILITY STATEMENT

This work is conducted using the open-source framework vLLM. We confirm that all hardware specifications and key hyperparameter details are reported in Sec. 4.2, dataset processing details are provided in Sec. A.2, and prompt settings are described in Sec. A.4.

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

# A   APPENDIX

## A.1   THE USE OF LLMS

We employed ChatGPT (OpenAI, 2025) and Gemini (Google, 2025) for the following purposes:

- polishing selected sentences to improve readability,
- exploring effective formats for figures and tables, and
- filtering useful requests in the ShareGPT dataset.

## A.2   DATASET DETAILS

**ShareGPT**: We use ChatGPT to filter useful requests and remove uninformative or nonsensical ones.

**TACO**: In thinking mode, we select requests containing fewer than 100 words, since excessively long requests are likely to produce generations exceeding 16,384 tokens, the default maximum length for all datasets in thinking mode.

## A.3   LLM GENERALIZATION

To further examine generalization, we also evaluate Qwen-32B and DeepSeek-V3.1, with Kendall's Tau values reported in Table 9. In the mathematics domain, both models perform nearly identically, suggesting comparable mathematical reasoning abilities. For code, Qwen-32B performs better, whereas DeepSeek-V3.1 shows a slight decline. In contrast, on general-domain tasks, DeepSeek-V3.1 achieves strong results. These findings indicate that our method remains effective across models of different scales and families.

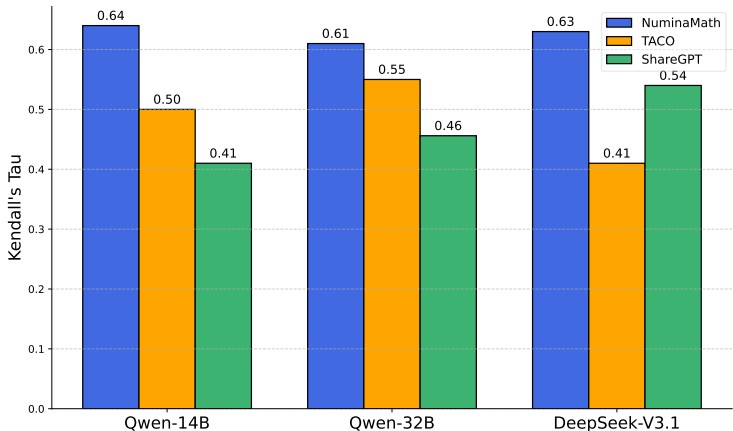

Figure 9: Kendall's Tau comparison of Qwen-14B, Qwen-32B, and DeepSeek-V3.1.

## A.4   PROMPT TEMPLATES

We provide math prompts for PREFILLONLY, RANKONLY, and LENGTHONLY. For the code and ShareGPT settings, the same templates are applied by simply replacing the math content with code or conversational tasks.

Math System Prompt (PrefillOnly)

```
CRITICAL FORMAT: Output ONLY a number, no extra text.
You are an expert at predicting response lengths and identifying the
    shortest query for math reasoning problems.
```

**Math Prompt (PrefillOnly)**

```
TASK: From the {mini_size} queries below, identify the SINGLE
    shortest one for math reasoning problems.
Return ONLY its position ID (0-9).

OUTPUT FORMAT:
<A NUMBER> (e.g. 5)

Problems:
Position 0: ...
Position 1: ...
```

**Math System Prompt (RankOnly)**

```
CRITICAL FORMAT: Output ONLY between markers, no extra text.
You are an expert in ranking math reasoning problems by difficulty.
Easier = faster = higher priority; harder = slower = lower priority.
```

**Math Prompt (RankOnly)**

```
TASK: Given {mini_size} problems, assign each a rank: 1=easiest  {
    mini_size}=hardest.
Ranks can repeat if difficulty is similar.

Difficulty guide:
1-3 easy, 4-6 moderate, 7-8 hard, 9+ hardest.

OUTPUT FORMAT:
<<
r1,r2,...,r{mini_size}
>>

Problems:
Position 0: ...
Position 1: ...
```

**Math System Prompt (LengthOnly)**

```
You are an expert at predicting response token counts for math
    reasoning problems.
Task: estimate exact token counts (not ranges).
Output only comma-separated numbers, no extra text.
```

**Math Prompt (LengthOnly)**

```
TASK: Predict token counts for {mini_size} math problems.

GUIDELINES:
- Simple: 80-200
- Standard: 200-400
- Complex: 400-700
- Very complex: 700+

OUTPUT FORMAT:
[number1],[number2],...,[number{mini_size}]
```

```
Problems:
Position 0: ...
Position 1: ...

Return only a single line:
n1,n2,...,n{mini_size}
```

## A.5 PERFORMANCE AT P99

For a more comprehensive analysis, we report the performance of TTFT, TPOT, and latency at P99 (Figure 10). The results show that improvements in latency and TPOT are even more substantial, highlighting the effectiveness of batching requests with similar lengths to mitigate long-tail effects and enhance parallel efficiency. However, this approach also introduces a drawback, as longer requests tend to have a negative impact on TTFT.

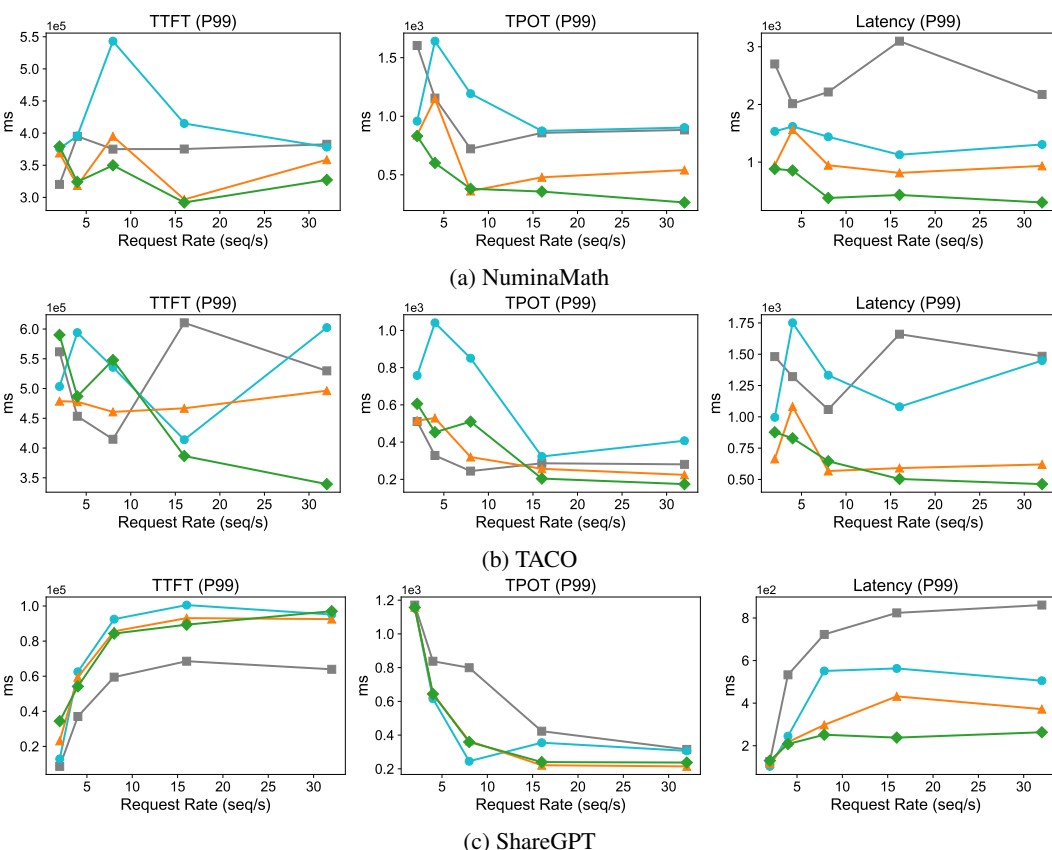

Figure 10: Performance of NuminaMath, TACO, and ShareGPT on TTFT, TPOT, and latency at P99.

## A.6 THROUGHPUT IN THINKING MODE

For a more comprehensive analysis, we evaluate throughput using 1,000 samples under two settings: a fixed duration of 20 minutes and a fixed workload of 100 requests. Table 3 presents the results under thinking mode. All datasets show clear improvements, consistent with the trends observed in the non-thinking setting.

| Dataset | Fixed Duration (20 minutes) | | | Fixed Workload (100 requests) | | |
|---|---|---|---|---|---|---|
| | **FCFS** | LENGTHONLY | **Speedup** | **FCFS** | LENGTHONLY | **Speedup** |
| NuminaMath | 25 | 124 | 4.96× | 108.23 min | 15.02 min | 7.21× |
| TACO | 7 | 22 | 3.14× | 301.17 min | 117.63 min | 2.55× |
| ShareGPT | 203 | 250 | 1.23× | 4.45 min | 2.80 min | 1.59× |

Table 3: Throughput comparison of scheduling methods across multiple domains in thinking mode.

