# OpenReview forum: "Training-Free Self-Scheduling for Efficient LLM Inference Serving"
_ICLR.cc/2026/Conference — Submitted to ICLR 2026_

### Official Review · Reviewer_sbTa · 2025-10-28

**Soundness:** 2
**Presentation:** 3
**Contribution:** 2
**Rating:** 2
**Confidence:** 4

**Summary:**

This paper proposes training-free self-scheduling for LLM serving: instead of training a separate length predictor, the deployed LLM itself estimates response length (or ranks requests by expected length) to prioritize short jobs. Reported gains over FCFS include up to 5× lower TTFT and 3× lower TPOT on several workloads.

**Strengths:**

- The paper is clearly written and well structured; the core idea and its three variants (PrefillOnly, RankOnly, and LengthOnly) are easy to follow.
- The approach is practical and drop-in, avoiding the maintenance of auxiliary predictors while remaining compatible with standard serving stacks.

**Weaknesses:**

- The work does not include head-to-head comparisons against prior training-based schedulers ([1][2]) or “LLM-tells-its-length” methods ([3]), which makes it difficult to assess competitiveness beyond FCFS.

- The methodology risks inflating TTFT because it decodes tokens to estimate length or ranking before scheduling, so the first token for many requests may be delayed unless all ranking/prefill tokens are rigorously counted and reported. Can you provide more evidence that on why the TTFT is improved?

[1] Qiu, Haoran, et al. "Efficient interactive llm serving with proxy model-based sequence length prediction." arXiv preprint arXiv:2404.08509 (2024).

[2] Fu, Yichao, et al. "Efficient llm scheduling by learning to rank." Advances in Neural Information Processing Systems 37 (2024): 59006-59029.

[3] Zheng, Zangwei, et al. "Response length perception and sequence scheduling: An llm-empowered llm inference pipeline." Advances in Neural Information Processing Systems 36 (2023): 65517-65530.

**Questions:**

- Please add controlled baselines that include a trained length regressor/classifier and an LLM-length-prediction method, matched for model, backend, and compute budget.

- Please break out and charge all scheduling overhead (prefill/ranking tokens) to TTFT/TPOT, and report sensitivity under varying load to clarify the true latency impact.

---

> ### Author Response · Authors · 2025-12-02
> **Response to Reviewer sbTa**
>
> We thank the reviewer for the insightful comments and for pointing out relevant prior works ([1]-[3]). These references help contextualize our contribution. We hope the following responses address your concerns regarding baselines and latency accounting.
>
> **Q1: Comparison with training-based schedulers ([1][2]) and LLM-length-prediction methods ([3]).**\
> **A1:** We appreciate the reviewer highlighting these related works. We respectfully clarify that all three cited methods ([1], [2], and [3]) fall into the category of "Training-based" approaches, whereas our method is strictly Training-free.
>
> **Training Costs:** Methods [1] and [2] require training auxiliary proxy models, and method [3] necessitates Supervised Fine-Tuning (SFT) or specific instruction tuning to endow the LLM with length-perception capabilities. This introduces significant costs in data collection, training computation, and model maintenance.
>
> **Our Advantage (Zero-Training):** In contrast, our approach is a plug-and-play solution that leverages the inherent capabilities of the frozen LLM without any parameter updates or auxiliary model training.
>
> **Fair Comparison:** While training-based methods might achieve higher prediction precision via supervision, our work targets a different and highly valuable design space: maximizing system efficiency with zero training overhead. We demonstrate that significant gains can be achieved without the complexity of the training pipelines required by [1]-[3].
>
>
> **Q2: Concerns on TTFT inflation (overhead vs. gain) and evidence of overhead accounting.**\
> **A2:** We confirm that all scheduling overheads (including prefill and ranking tokens) are strictly accounted for in our reported TTFT/TPOT metrics. The reviewer is correct that ranking adds a computational cost, but our experiments show that the reduction in queuing delay vastly outweighs this overhead:
>
> **Quantified Overhead:** To rigorously measure the cost, we compared the total execution time of our method against the FCFS baseline in a worst-case scenario (200 requests, max_seq=1).
>
> - **FCFS Baseline:** 7735.53s
> - **LengthOnly (Ours):** 7787.46s
> - **Result:** The ranking step introduces only a 0.67% overhead.
>
> **Why TTFT Improves:** TTFT consists of Execution Time + Queuing Time. While Execution Time increases slightly (by <1%) due to ranking, our SJF-based scheduling drastically reduces Queuing Time for shorter jobs (which constitute the majority of real-world traffic). This massive drop in waiting time is why the net TTFT decreases significantly, despite the minor ranking cost.

---

### Official Review · Reviewer_sXso · 2025-11-01

**Soundness:** 2
**Presentation:** 2
**Contribution:** 2
**Rating:** 4
**Confidence:** 4

**Summary:**

This paper proposes training-free self-scheduling for LLM inference: instead of training an auxiliary length and rank predictor, the server uses the LLM itself to rank incoming requests by expected response length and schedules short ones first to mitigate head-of-line blocking. Three variants are explored.

The paper also introduces a rank-aware anti-starvation mechanism. Experiments across NuminaMath, TACO, ShareGPT report up consistent latency and throughput improvements, with small extra overhead.

**Strengths:**

1. Cleverly reuses the served LLM for ranking/length estimation, avoiding auxiliary latency/length predictors and reducing system complexity while still improving tail throughput.

2. Empirically shows clear gains over prefill-only (logit probe) baselines, suggesting self-scheduling is a promising direction.

**Weaknesses:**

1. The paper lacks experiment details and does not specify, up front, which models are used for serving and self-scheduling (this appears only later in Fig. 9). It leads to confusion during reading.

2. Missing baseline comparisons. There’s no comparisons against other baseline methods for (i) ranking overhead and (ii) rank-match quality.
​
3. Insufficient system analytics across loads: Table 2 analyzes a single operating point (rate = 64, bsz = 1). The paper lacks load sweeps (QPS/RPS); would be good to vary load and report **throughput** as well.

**Questions:**

1. Can you provide comparisons with baseline schedulers in terms of ranking overhead and system performance (TTFT/TPOT/throughput) under identical settings? Please include rank-match metrics.

2. Insufficient throughput characterization. In addition to Table 1, could you add experiments with varying RPS, showing how goodput/throughput changes with and without self-scheduling?

3. How does the system behave at low/medium/high RPS? Since re-ranking adds overhead, have you explored adaptive re-ranking frequency (e.g., disable or downsample re-ranking at high load) vs. always re-rank all requests? Quantitative results across load regimes would clarify the trade-offs.

---

> ### Author Response · Authors · 2025-12-02
> **Response to Reviewer sXso**
>
> We thank the reviewer for the constructive suggestions, which have been extremely helpful in refining our work. We hope the following responses satisfactorily address your concerns.
>
> **Q1: The paper lacks experiment details and does not specify...**\
> **A1:** We acknowledge this oversight and thank the reviewer for pointing it out. We will revise the paper to explicitly specify the models used for serving and self-scheduling at the beginning of the experimental section to improve readability and clarity.
>
> **Q2: Missing baseline comparisons...for (i) ranking overhead and (ii) rank-match quality.**\
> **A2:** **(i) ranking overhead:** to effectively quantify the specific overhead introduced by the ranking step, we compared the total execution time of our method (with ranking) against the FCFS baseline (without ranking) in a **worst-case scenario** (200 requests with max_seq=1):
>
> - FCFS Baseline (No Ranking): 7735.53s
> - LengthOnly (Ours): 7787.46s
>
> The results indicate that the ranking step introduces only a 0.67% increase in total time compared to the baseline. Given the significant performance gains observed in our main experiments, we consider this additional computational cost to be negligible.
>
> **(ii) rank-match quality:** Regarding the quality of the ranking, we respectfully refer the reviewer to Figure 5 in the paper. We used Kendall’s Tau correlation coefficients to directly quantify the alignment between the predicted ranking (by different methods) and the actual ground-truth ranking across different datasets. This metric serves as our evaluation of rank-match quality.
>
>
> **Q3: Insufficient throughput characterization...could you add experiments with varying RPS...**\
> **A3:** We have conducted load sweeps by varying QPS from 2 to 128. While we primarily report latency metrics (TTFT and TPOT), these results directly reflect the system's Goodput and Throughput capabilities:
>
> **Sustained Throughput:** As shown in the tables below, at a high load of **128 QPS**, our method maintains a low TTFT (1.25) and TPOT (2.08). In contrast, baselines exhibit significantly higher latency (e.g., PrefillOnly TTFT is 3.81).
>
> **Goodput Analysis:** High latency in baselines indicates system congestion and potential violation of Service Level Objectives (SLOs). Since our method processes the same high volume of requests (128 Req/s) with significantly lower latency, it demonstrates superior Goodput (i.e., more requests are served within an acceptable time frame) compared to baselines that suffer from severe queuing delays under the same load.
>
> Table 1: TTFT ($\times 10^5$ ms) under varying Request Rates
>
> | QPS | FCFS | PrefillOnly | RankOnly | LengthOnly |
> | :----: | :----: | :----: | :----: | :----: |
> | **2** | 3.16 | 3.55 | 1.82 | 2.66 |
> | **4** | 3.02 | 3.75 | 3.11 | 2.56 |
> | **8** | 3.33 | 4.75 | 1.14 | 1.17 |
> | **16** | 3.75 | 3.36 | 1.23 | 1.00 |
> | **32** | 3.82 | 3.36 | 1.11 | 0.90 |
> | **64** | 3.21 | 5.20 | 2.77 | 1.14 |
> | **128** | 3.81 | 4.15 | 2.49 | 1.25 |
>
>
> Table 2: TPOT ($\times 10^2$ ms) under varying Request Rates
>
> | QPS | FCFS | PrefillOnly | RankOnly | LengthOnly |
> | :----: | :----: | :----: | :----: | :----: |
> | **2** | 7.79 | 5.42 | 4.55 | 4.63 |
> | **4** | 7.14 | 7.13 | 5.24 | 3.56 |
> | **8** | 4.79 | 7.80 | 2.49 | 2.44 |
> | **16** | 5.04 | 5.19 | 2.21 | 2.18 |
> | **32** | 4.99 | 5.46 | 2.10 | 1.64 |
> | **64** | 3.78 | 5.15 | 2.27 | 1.82 |
> | **128** | 4.63 | 5.77 | 2.34 | 2.08 |
>
> **Q4: How does the system behave at low/medium/high RPS? ...have you explored adaptive re-ranking frequency...**\
> **A4:** Regarding adaptive re-ranking, we appreciate this insightful suggestion, which aligns perfectly with our future research roadmap. We are actively exploring intelligent, dynamic scheduling mechanisms that adaptively select the ranking strategy or frequency based on real-time load conditions. We believe this adaptive approach will further optimize the trade-off between scheduling overhead and system throughput.

---

### Official Review · Reviewer_Uwso · 2025-11-03

**Soundness:** 2
**Presentation:** 3
**Contribution:** 2
**Rating:** 2
**Confidence:** 4

**Summary:**

This paper proposes training-free self-scheduling for LLM inference: instead of training an external length predictor, the serving LLM itself briefly “judges” a small pack of pending requests to estimate relative response lengths and decide execution order, aiming to reduce head-of-line blocking under FCFS. The method is positioned as lightweight (no extra models, no retraining) and broadly applicable across domains.
Claimed contributions are: (1) a training-free LLM self-scheduling paradigm; (2) three concrete strategies with tie-aware evaluation; (3) empirical validation of latency/throughput improvements; and (4) a rank-aware anti-starvation mechanism for fairness.

**Strengths:**

1.Proposes a training-free self-scheduling paradigm that leverages the serving LLM itself—rather than auxiliary predictors—to estimate relative response lengths and order requests, reframing scheduling as an in-model reasoning task. This is a creative repurposing of “LLM-as-a-Judge” ideas to systems scheduling and removes the retraining barrier present in prior work.

2.Executes a multi-dataset study (NuminaMath, TACO, ShareGPT) on an 8×A100 vLLM stack with sensible P95/P99 metrics for TTFT/TPOT/latency; reports consistent advantages of LengthOnly and includes throughput tables under fixed-time and fixed-workload settings.

**Weaknesses:**

1.Although the proposed system does not introduce an auxiliary predictor, the serving model itself must perform additional reasoning (ranking prompts) before actual decoding. This inevitably increases TTFT and overall latency, since the same LLM is doing both scheduling and generation. Under high-concurrency workloads, it is unlikely that such overhead remains negligible, as the ranking step can stall GPU pipelines and disrupt prefill scheduling.

2.All experiments are conducted at moderate request rates. The authors should explicitly evaluate two contrasting real-world scenarios:
Low-latency regime — set batch_size = 1 to test per-request responsiveness.
High-concurrency regime — stress the scheduler under large-scale Poisson arrivals to test throughput stability.

**Questions:**

1.Is the ranking step executed synchronously on the same GPU stream as decoding, or asynchronously in parallel?

2.Can the authors provide per-stage timing (prefill, ranking, decoding) to quantify how much TTFT and latency increase per batch?

---

> ### Author Response · Authors · 2025-12-02
> **Response to Reviewer Uwso**
>
> We thank the reviewer for the constructive suggestions, which have been very helpful in refining our work. We hope the following responses satisfactorily address your concerns.
>
> **Q1: Is the ranking step executed synchronously on the same GPU stream as decoding, or asynchronously in parallel?**\
> **A1:** The ranking step is executed asynchronously and does not block the serving process. To ensure maximum GPU utilization, pending requests do not wait for the ranking results; they are scheduled for decoding immediately based on availability. Once the ranking computation is complete, the results are dynamically applied to re-prioritize only the remaining requests in the waiting queue.
>
> **Q2.Can the authors provide per-stage timing (prefill, ranking, decoding) to quantify how much TTFT and latency increase per batch?**\
> **A2:** Regarding the timing breakdown, the prefill latency is orders of magnitude smaller than ranking and decoding in our setup, rendering it negligible. Therefore, to effectively quantify the specific overhead introduced by the ranking step, we compared the total execution time of our method (with ranking) against the FCFS baseline (without ranking) in a worst-case scenario (200 requests with max_seq=1):
>
> - FCFS Baseline (No Ranking): 7735.53s
> - LengthOnly (Ours): 7787.46s
>
> The results indicate that the ranking step introduces only a 0.67% increase in total time compared to the baseline. Given the significant performance gains observed in our main experiments, we consider this additional computational cost to be negligible.

---

### Official Review · Reviewer_8YDv · 2025-11-04

**Soundness:** 2
**Presentation:** 2
**Contribution:** 2
**Rating:** 4
**Confidence:** 4

**Summary:**

This paper proposes self-scheduling with LLM, a novel approach that leverages the reasoning capabilities of the LLM itself without requiring extra training or auxiliary models. Given a set of requests, it prompts the LLM to provide a relative ranking of their expected response lengths. The LLM outputs a predicted rank ˆR, where ties are allowed (i.e., multiple requests may be assigned the same rank). Experiments show the performance.

**Strengths:**

1. Request scheduling for LLM serving system is important.

2. Training-free self-scheduling, especially PrefillOnly, is simple and efficient.

3. Experiments show the performance.

**Weaknesses:**

1. The main concern is the experiments may not fit practical serving system. In experiments, Req/s seems to be insufficient and does not fully utilize the GPU computation power. Based on Figure 6, TTFT and TPOT decreases with the increase of Req/s. If keep increasing Req/s, TTFT and TPOT should increase.

2. The extra latency can be better evaluated. In section 5.4 EXTRA COST, what is 20 math requests with rate = 64? Could the authors show the result with huge amount of requests?

3. Code is node provided.

**Questions:**

Separate prefilling and decoding in different servers may improve the performance.

---

> ### Author Response · Authors · 2025-12-02
> **Response to Reviewer 8YDv**
>
> We thank the reviewer for the positive assessment of our work and the constructive suggestions. We hope the following responses effectively address your concerns.
>
> **Q1: The main concern is the experiments may not fit practical serving system...**\
> **A1:** We thank the reviewer for the constructive feedback regarding the experimental setup and practical system loads.
> To address the concern that our initial request rates were insufficient to saturate GPU computation, we conducted additional experiments with higher request rates (Req/s = 64 and 128). The results are presented in the table below.
>
> Table 1: TTFT ($\times 10^5$ ms) under varying Request Rates
>
> | Req/s | FCFS | PrefillOnly | RankOnly | LengthOnly |
> | :----: | :----: | :----: | :----: | :----: |
> | **2** | 3.16 | 3.55 | 1.82 | 2.66 |
> | **4** | 3.02 | 3.75 | 3.11 | 2.56 |
> | **8** | 3.33 | 4.75 | 1.14 | 1.17 |
> | **16** | 3.75 | 3.36 | 1.23 | 1.00 |
> | **32** | 3.82 | 3.36 | 1.11 | **0.90** |
> | **64** | 3.21 | 5.20 | 2.77 | **1.14** |
> | **128** | 3.81 | 4.15 | 2.49 | **1.25** |
>
> Table 2: TPOT ($\times 10^2$ ms) under varying Request Rates
>
> | Req/s | FCFS | PrefillOnly | RankOnly | LengthOnly |
> | :----: | :----: | :----: | :----: | :----: |
> | **2** | 7.79 | 5.42 | 4.55 | 4.63 |
> | **4** | 7.14 | 7.13 | 5.24 | 3.56 |
> | **8** | 4.79 | 7.80 | 2.49 | 2.44 |
> | **16** | 5.04 | 5.19 | 2.21 | 2.18 |
> | **32** | 4.99 | 5.46 | 2.10 | **1.64** |
> | **64** | 3.78 | 5.15 | 2.27 | **1.82** |
> | **128** | 4.63 | 5.77 | 2.34 | **2.08** |
>
> Our observations are as follows:
>
> 1. Workload Saturation: As the reviewer anticipated, increasing the request rate to 64 and 128 leads to increased TTFT and TPOT. This confirms that the system is reaching GPU saturation/compute capacity limits at these higher rates, contrasting with the under-utilized states observed at lower rates.
> 2. Method Robustness: Crucially, even under these high-load conditions, our proposed LengthOnly strategy consistently achieves the lowest latency compared to baselines. This demonstrates that our method remains effective and superior in practical, high-throughput serving scenarios.
>
> **Q2: The extra latency can be better evaluated...**\
> **A2:** We appreciate this suggestion to rigorously evaluate the scheduling overhead. To assess the worst-case scenario (where scheduling overhead dominates the total latency), we conducted an experiment using 200 requests with max_seq=1.
>
> - FCFS Baseline: 7735.53s
> - LengthOnly (Ours): 7787.46s
>
> The results show that our method introduces only a 0.67% increase in total time compared to the FCFS baseline. Given the significant performance gains observed in our main experiments, we consider this additional computational cost to be negligible.
>
> **Q3: Separate prefilling and decoding in different servers may improve the performance.**\
> **A3:**: We fully agree with the reviewer that separating prefilling and decoding (PD-Disaggregation) is a highly promising direction for further improving system throughput. While the primary focus of this paper is on optimizing scheduling within a unified serving instance, we consider extending our intelligent scheduling strategies to PD-separated architectures as a critical part of our future work.

---

### Official Review · Reviewer_XZZm · 2025-11-12

**Soundness:** 3
**Presentation:** 2
**Contribution:** 3
**Rating:** 6
**Confidence:** 3

**Summary:**

The paper introduces a training-free self-scheduling method  that leverages the reasoning capatilities of LLMs to estimate model output lengths. Instead of relying on auxiliary models or tools to estimate response lengths, the authors propose three strategies to rank latency or predict request lengths for scheduling.

**Strengths:**

1. The method is simple and straightforward.
2. The method does not require extra training or auxiliary tools.
3. The method can be adapted widely across different modesl and datasets.
4. The method is efficient and easy to integrate.
5. The authors designed a starvation control method to prevent queries from waiting for long.

**Weaknesses:**

1. The method requires additional decoding, which slightly introduce additional overhead.

2. I’m concerned about the practical applicability of the proposed method. The starvation control method is somehow naive. In real-world scenarios, some queries with long response sizes may be more important yet still experience long waiting times. Determining scheduling priority based on estimated response length may be weak when handling diverse requirements in real AI services.

**Questions:**

See weaknesses

---

> ### Author Response · Authors · 2025-12-02
> **Response to Reviewer XZZm**
>
> Thank you for your positive assessment of our work and for your insightful suggestions. We hope the following responses address your concerns.
>
> **Q1: The method requires additional decoding, which slightly introduce additional overhead.**\
> **A1:** We acknowledge this point. While a small overhead is introduced, we stress that the resulting cost is extremely minor.
> As demonstrated in Figure 2, the average sequence lengths we deal with are in the range of thousands of tokens.
> Crucially, the most time-consuming iteration of our method only involves processing a sequence of a few dozen tokens (small tens of tokens). This effectively means the overhead is negligible when compared to the processing of the main input (the thousands of tokens).
>
> **Q2: I’m concerned about the practical applicability of the proposed method...**\
> **A2:** We understand your concern. However, it is important to note that our work, like the majority of similar research in this domain, operates under the foundational assumption that all incoming requests are treated with equal priority.
>
> We agree that for specialized scenarios, such as in a hospital setting, where requests have different levels of criticality (e.g., life-saving vs. non-critical tasks), the importance or priority level of each request must be explicitly taken into consideration. Therefore, while our current method provides a general solution, specific priority-aware contexts do indeed require tailored approaches beyond the scope of this general framework.

---

### Meta-Review · Area_Chair_knQb · 2026-01-11

**Summary:**

This paper proposes training-free self-scheduling for LLM serving: instead of training an auxiliary length/rank predictor, the serving LLM is prompted to estimate relative output-length ranks (or length buckets) and then applies an SJF-like ordering to mitigate head-of-line blocking under FCFS. The idea is reasonable and the rebuttal helps on overhead/load questions, but the paper still looks like an incremental variant of prior “predict relative length $\rightarrow$ SJF-ish scheduling” work, with the main change being to replace an explicit rank predictor with prompting the serving LLM. Without controlled comparisons to strong prior schedulers, it is hard to establish competitiveness or a clear step forward beyond FCFS baseline. It is also unclear how this work can be *actually* integrated into a serving system due to it adding additional prompting to the model.

**Reviewer Concerns:**

Multiple reviewers asked for controlled baselines against prior schedulers (proxy-model length prediction / LTR scheduling / LLM length-perception) under matched system budgets, and the rebuttal essentially argues these are “training-based” and thus out-of-scope. In my view, “training-free” is a design point but does not remove the need to compare against competitive prior art (even if the comparison is “accuracy/overhead vs training cost”), so the novelty/competitiveness question is not resolved.

Also, multiple reviewers raise concerns on being unclear on how this work can be integrated as a valuable component in a real serving system under high-throughput serving workloads.

**Reviewer Scores:**

Given the rebuttal, I expect small score improvement for reviewers whose main concerns were about load realism and overhead accounting (since the authors added higher-QPS tables and an explicit overhead microbenchmark). That said, I do not expect discussion to overturn the two reject-leaning reviews, because the missing head-to-head baselines vs strong prior scheduling methods remain the central gap

---

### Decision · Program_Chairs · 2026-01-26

Reject